# Policy and behavioral response to shock events: An agent-based model of the effectiveness and equity of policy design features

**Vivek Shastry**[1]*, **D. Cale Reeves**[1,2], **Nicholas Willems**[3], **Varun Rai**[1,3]

**1** LBJ School of Public Affairs, The University of Texas at Austin, Austin, TX, United States of America,
**2** School of Public Policy, Georgia Institute of Technology, Austin, TX, United States of America,
**3** Department of Mechanical Engineering, The University of Texas at Austin, Austin, TX, United States of America

* svivekshastry@utexas.edu

## Abstract

In the aftermath of shock events, policy responses tend to be crafted under significant time constraints and high levels of uncertainty. The extent to which individuals comply with different policy designs can further influence how effective the policy responses are and how equitably their impacts are distributed in the population. Tools which allow policymakers to model different crisis trajectories, policy responses, and behavioral scenarios *ex ante* can provide crucial timely support in the decision-making process. Set in the context of COVID-19 shelter in place policies, in this paper we present the COVID-19 Policy Evaluation (CoPE) tool, which is an agent-based modeling framework that enables researchers and policymakers to anticipate the relative impacts of policy decisions. Specifically, this framework illuminates the extent to which policy design features and behavioral responsiveness influence the efficacy and equity of policy responses to shock events. We show that while an early policy response can be highly effective, the impact of the timing is moderated by other aspects of policy design such as duration and targeting of the policy, as well as societal aspects such as trust and compliance among the population. More importantly, we show that even policies that are more effective overall can have disproportionate impacts on vulnerable populations. By disaggregating the impact of different policy design elements on different population groups, we provide an additional tool for policymakers to use in the design of targeted strategies for disproportionately affected populations.

## 1. Introduction

Shock events–including crises such as the COVID-19 pandemic–have three common characteristics–a *threat* to shared values, a sense of *urgency*, and high degrees of *uncertainty* [1]. In the aftermath of shock events, policymakers often make rapid decisions under uncertain conditions. Uncertainty can arise from at least three sources–the problem, the response, and the

**Data Availability Statement:** All related code and data are available in a public GitHub repository. The citation and link to the repository is mentioned below. Rai Group. (2020). CoPE: COVID-19 Policy

Evaluation tool (0.1) [Computer software]. https://github.com/RaiResearchGroup/CoPE.

**Funding:** The authors received no specific funding for this work.

**Competing interests:** The authors have declared that no competing interests exist.

public reaction to the response [1]. The first source of uncertainty comes from difficulties in estimating potential trajectories of an evolving crisis. The second source of uncertainty is in the design of the policy response, such as the scope, timing, duration, and targeting of the policy. The third source of uncertainty arises from how people respond to the policies, such as the willingness of individuals to comply with directives, depending on their perception of risk of non-compliance. Such individual behavioral responses can constrain or amplify the overall effectiveness of the policy response, as well as how equitably the policy impacts are distributed among different population groups.

In the COVID-19 pandemic case, countries rapidly implemented a wide range of strategies in the face of very limited information and high uncertainties. One of the lessons emerging from this crisis is that there is often no single best policy response, and it is important to have a wide range of response strategies available to be implemented depending on the context [2]. Given the uncertainties and limited resources that policymakers have at their disposal, decision tools that allow them to model different crisis trajectories, policy responses, and behavioral scenarios *ex ante* can provide timely and crucial support in the decision-making process.

Set in the context of COVID-19 shelter in place policies, in this paper we present the COVID-19 Policy Evaluation (CoPE) tool, which is an agent-based modeling framework that enables researchers and policymakers to anticipate the relative impacts of policy decisions. CoPE is a hybrid tool combining an agent-based model that generates dynamic social and professional contact networks using geographically specific census data with an epidemiological model of the progression of the COVID-19 disease through agents. The interactions that generate exposures are partly governed by the policy context which varies based on the duration, timing, and targeting of a shelter in place (SIP) policy, and partly governed by the extent to which individuals comply with the policy. The objective of this paper is to demonstrate the application of the CoPE tool and illuminate the extent to which policy design configurations and behavioral responsiveness influence the efficacy and equity of policy responses to shock events. One *ex ante* expectation for SIP policy is that early enactment, longer durations, and fewer essential workers would be generally more effective in reducing the transmissions and subsequent hospitalizations. Through our analysis, we show that different combinations of policy design features produce synergies and tradeoffs, especially when it comes to the equity of policy outcomes.

The rest of this paper is structured as follows. We review the relevant background literature in section 2, detail our data and methods in section 3, discuss the results in section 4 and conclude in section 5.

## 2. Background

Four categories of core ideas are discussed herein–policy design, behavioral response, outcome evaluation, and modeling approach. First, we discuss how policymakers implement a wide range of policy designs during shock events in general and COVID-19 in particular, often with limited ability to understand and anticipate the potential impacts on these policies on the local communities and across socio-demographics. Second, we focus on why individuals respond differently to policy directives and how these predispositions influence their compliance specifically vis-à-vis COVID-19 shelter in place policies. Third, we highlight the evaluation of the distributional equity of policy outcomes *before* the policy is actually implemented, an important but often challenging endeavor. Finally, we compare two approaches to epidemiological modeling in terms of their ability to generate this desirable output. Together, we develop and make a case for deploying tools that can help policymakers model the suite of policy design options and individual behavioral feedback mechanisms together, with the objective of facilitating an *ex-ante* evaluation of the effectiveness *and* equity of these policy designs.

## 2.1. Policy response to shock events

Shock events, often in the form of disasters, wars, and economic or public health crises can induce policy change [3,4]. The *multiple streams theory* of the policy process suggests that an issue can receive serious attention from decision makers when *policy entrepreneurs* are able to bring recognition to *problems* and provide viable *policy* solutions that are *politically* aligned at an opportune time (*window of opportunity*) [5]. Ordinarily, the development of policy alternatives is left to experts who can analyze successful past policies and provide appropriate options for the current context. Some windows of opportunity can be expected (for example budget cycles or elections), which gives policy entrepreneurs sufficient time to analyze policy options to be presented to the decision maker at the right time [6].

In contrast, shock events or crises present unexpected windows of opportunity, often when there is no time to fully anticipate the complexity of problems and the heterogeneous impacts of policy responses. An appropriate and timely policy response in the face of a crisis requires the policymakers to both maintain a level of surveillance over emerging threats as well as have tailored policy options ready for when the crisis hits [7]. In rapidly unfolding, complicated situations, failure to consider alternative options, poor information search, selection bias in processing information and failure to examine costs and risks of preferred choices can lead to defective policymaking [8]. In such cases, new tools are necessary for rapidly generating context-specific scenarios, allowing for *ex-ante* evaluations of policy responses that are tailored to local conditions [9,10].

The COVID-19 crisis has resulted in the implementation of a myriad of local responses that seldom demonstrate an adequate understanding of the potential impacts of policy responses on different population groups [11]. Policy instruments that have been used to contain the spread of COVID-19 and other infectious diseases include social distancing requirements, mask wearing guidelines, shelter in place orders, testing and contract tracing strategies, domestic and international travel restrictions, school closures, etc. [12,13]. A common policy instrument used by local policymakers is the shelter in place order (also sometimes referred to as Stay at home order or lockdown, hereafter abbreviated SIP). Policy design features of SIP orders include the extent to which professional and social activities remain open, social distancing, mask requirements, capacity limits and other guidelines.

In the United States, in the absence of uniform federal mandates, local and state administrations bore the responsibility for developing their own response strategies. Local policymakers enacted a range of SIP orders, with broad variation in how quickly SIP was enacted, how long SIP was enforced, and who were classified as essential workers [14]. For example, SIP started as early as March 17[th] in some California counties and two weeks later in states like Texas and Florida. The orders were in place for only a month in Texas and Florida and for two months or longer in Washington and Michigan. The heterogeneity in policy responses have been shown to be associated with uncertainties in the understanding of the disease transmission parameters as well as uncertainties about potential downstream economic costs of SIP orders [15]. The effectiveness of different SIP policy design features is still an open question.

## 2.2. Individual behavioral heterogeneity

One of the sources of uncertainty in predicting policy outcomes, as mentioned in section 1, arises from attempting to anticipate the extent to which people comply with policy directives. Population level compliance is an aggregation of individual abilities and decisions to comply, and human beings are boundedly rational decision makers constrained by limited information, behavioral habits, cognitive associations, time, risk perception, uncertainty and a range of other factors [16–19]. During epidemics, as during other crises, a significant psychological

burden is placed on individuals which impacts their behavior. Epstein et al. introduced the idea of a "coupled contagion" which suggests that individuals react not only to the disease itself, but also to the fear of the disease. When individuals are "infected" with fear, they self-isolate and take themselves out of circulation. However, once they are "cured" of this fear, they can get back into circulation and contribute to the subsequent waves of infections [20].

Threat perception and social context are important among the many behavioral pathways that affects how individuals make decisions [21]. While a strong threat perception can evoke protective behavior from individuals, an "optimism bias" [22]–which makes an individual feel they are less likely to contract the disease than others–can make them ignore their threat perceptions. An early study of COVID-19 in the United States found evidence of this optimism bias noting that while people generally increased protective behaviors, a sub-group of individuals who felt that the epidemic would not affect them personally reported low engagement in protective behaviors [23].

The heterogeneity in risk perception among individuals covaries with many factors. Risk perception has been shown to be dependent on the prevalence of disease in an individual's social network [24]. Risk perception and protective behavior also vary among different demographic groups, where older individuals, females, or individuals with higher educational level are more likely to engage in protective behavior [25,26]. The relationship of risk perception with age–a monotonic decrease [27]–is consistent with a number of media reports since the beginning of COVID-19 which have indicated that younger individuals have tended to ignore public health warnings and engage in risky behavior. Risk perception is an important behavioral response that affects individual compliance with local policies, and modeling efforts should factor in these dynamics.

## 2.3. Evaluation of outcomes

Substantial global variations exist in government policy responses to COVID-19 [13]. These policy responses have been evaluated with regard to their overall effectiveness in reducing mobility [28], healthcare demand [12] and hospitalizations [29]. Specifically focusing on the evaluation of SIP policies in the United States, states and counties that adopted SIP policies early on were found to benefit the most from their policies in terms of reduced number of cases [30–32]. While most of these studies focus on the timing of the SIP policy enactment, there are other elements of a SIP policy design that deserve attention, such as the duration and the distinction between occupations that must discontinue working and those that can continue. While it may be reasonable to expect that a longer duration SIP policy and those that target more occupations may be generally more effective, forming expectations about the relative magnitudes of impact from these policy design changes is less straight forward.

Additionally, the distributional equity of the policy impacts is an important dimension to investigate. During crises, local decision makers face high uncertainties about enacting, delaying, targeting or terminating policy responses, and the economic and social inequalities that ensue from these decisions have been understudied in the policy sciences [33]. In the context of rising social inequalities, analyses of distributional impacts of public policy decisions are being undertaken across many fields [34–36]. Focusing on COVID-19, there is now growing evidence of disparities in the rates of infection among racial and socio economic groups, with the more disadvantaged population groups facing a higher burden of infection [37,38]. Many of these disadvantaged groups already face underlying systemic barriers such as lower access to healthcare and higher prevalence of chronic health conditions that further aggravate the burden faced by these communities due to COVID-19 [39–41]. Higher infection rates have been found among essential workers who were not able to reduce their mobility as much

as others, and instead worked in densely packed places where the risk of infection is higher [37].

The ability of individuals to comply with policy directives can alter how they are impacted by the epidemic [42]. In addition to individual risk perception, compliance to policies is also constrained by individual socio-economic conditions. Emerging research shows that workers in occupations that have low probability of being able to work from home are also more economically and socially vulnerable and face higher risks of infection [43,44]. Residents in economically impoverished neighborhoods are less likely to comply with SIP orders, for several reasons including being employed in essential jobs, having to travel farther distances to access services and living in dense environments that makes social distancing difficult [11,42,45]. Compliance can also change over time due to changes in risk perceptions or changes in socio-economic conditions as a result of social safety net programs. A poorly conceived policy response can aggravate the inequities and cause irreparable damage to the fabric of these communities [46]. In addition to evaluation of overall effectiveness of SIP and other mitigation policies, understanding the distributional equity of impacts must be at the center of analysis. To that end, we present a decision tool for simulating the interactions of policy and behavioral responses and comparing the impact of SIP design features on the effectiveness and distributional equity of COVID-19 outcomes. This tool is based on an agent-based modeling framework, which is discussed next.

## 2.4. Modeling approaches

Two broad approaches to epidemiological modeling are those rooted in a compartmental approach and the agent-based approach. Compartmental models, for example the SIR (Susceptible-Infected-Recovered) model and variations of it, are commonly used to model the transmission of infectious diseases [47–49]. These models have been recently used in the United States to track the anticipated impacts of COVID-19 policy configurations [50,51]. They provide aggregate outcomes of policy effectiveness and have contributed significantly to the policy responses to the COVID-19 epidemic. These models work by applying a set of differential equations that govern disease progression to a pre-defined population matrix. Because compartmental models do not have highly resolved individuals as the primary unit of analysis, they have difficulty analyzing individual level impacts or factoring in dynamic behavior of individuals in the course of the analysis. Both these aspects are key to understanding the distributional equity of policy responses. While they come with their own strengths and limitations, agent-based models (ABMs) offer an approach to overcome the fundamental limitation of the compartmental models describe above.

At the core of ABMs are heterogeneous individual *agents* who interact with each-other and their *environment* according to a set of behavioral *rules* [52]. This approach allows for the specification of different policy configurations and dynamic decision-making criteria, such as deciding to be more cautious or flout the rules, to study how macro-level outcomes emerge from individual-level decisions over time. When the characteristics of each agent (e.g. income, occupation class, age, etc.) are well-resolved, ABMs open the possibility of exploring distributional effects of different policy specifications. ABMs incorporate uncertainty by introducing stochastic agent interactions and by simulating multiple runs for each input scenario [53].

ABMs have been used to explore behavioral processes and equity of outcomes in technology adoption [54], rates of incarceration [55], emergency responses [56], access to healthy foods [57], agriculture [58], and many other fields [59,60]. ABMs have also been used in the simulation of previous epidemics such as smallpox [61], H5N1 influenza [62] and H1N1 influenza [63,64]. In the context of COVID-19 policy design features, ABMs have been developed to

explore the overall effect of travel restrictions [65], testing and contact tracing [66], and timing and duration of social distancing measures [67,68]. Some of these studies also highlight the importance of population level compliance to the social distancing guidelines [66,67]. We contribute to these ongoing efforts by adding the ability to simulate several policy design features interactively along with dynamic and empirically grounded behavioral responses from the agents. Further, we focus our analysis on understanding the distributional equity of the different policy responses, in addition to their overall effectiveness.

## 3. Data and methods

The decision tool we present in this paper, COVID-19 Policy Evaluation (CoPE), is a flexible, modular, empirically and epidemiologically grounded Agent-Based Model (ABM). Here we elaborate on the key components of the agent-based CoPE tool, including the input data, the operational modules that drive the model, the research design we employ, and our analytical strategy. The model code and user guides have been made publicly available on GitHub [69], and additional information on the key parameters and equations used in the model are provided in the supplementary information file (see S1 Appendix).

### 3.1. Input data

CoPE leverages household level demographic data from the American Community Survey (ACS) [70] and can therefore model context specific population-scale scenarios for any location in the United States. This is important, because the demographic and occupational composition of every location is different, and therefore the impact of identical policy configurations might result in different impacts in different locations. To generate household level characteristics, we leverage several variables from the most recent (2019) American Community Survey (ACS) 5-year estimates. The lowest resolution that ACS captures these variables is at a block-group level. Using the per-block-group number of households by race/ethnicity (variable groups B11001B-I) paired with the joint distribution of age, income, and race/ethnicity at the census track level (B19037B-I) we estimate the same joint distribution at the block group level. Then, by block group, we merge household size (B11016) and occupation (C24010) features.

   To summarize, CoPE takes the census code for any geographic unit in the U.S. as an input parameter and implements the model using the household distributions for all block groups in that geographic unit. The result is a population model that can probabilistically generate agents with age, race/ethnicity, income, occupation, and household size attributes matched at the block-group level for any given location (city, county, metropolitan region, or state) in the United States. For the purpose of our analysis presented in this paper, we generate household distributions for all block-groups in Travis County.

### 3.2. Operational modules

CoPE consists of three key modules that together generate the emergent outcomes we analyze here. The first module sets the *policy context* through three policy decisions related to SIP policy: timing, duration, and targeting. The second module determines *inter-agent interactions*–building up the potential for exposure among agents. The third module, *epidemiological progression*, charts each agent's path through the disease from exposure, potential hospitalization, and eventual recovery or death. CoPE is therefore a hybrid tool which integrates a compartmental model within the agent-based model. Feedback loops connect each of the three modules, allowing agents to change their behavior in response to the current policy context, the infection states of their social and professional connections, and their own infection status. We

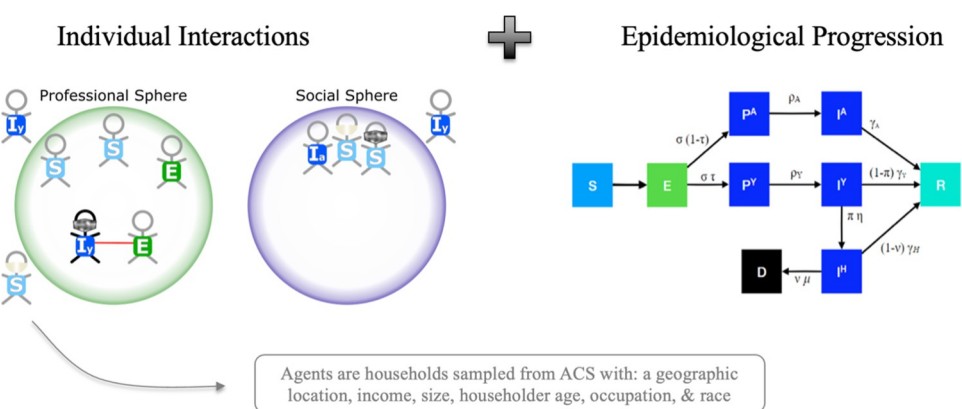

**Fig 1. Exposure and progression dynamics.** CoPE is a bottom-up model of the way individuals interact with one another both in a social sphere and in a professional sphere. Professional sphere includes on-the-job essential interactions, as well as non-job essential interactions such as grocery shopping. These interactions therefore capture exposure dynamics that are sensitive to policy such as which occupation is designated as an essential service. This exposure generating model is overlaid on the compartmental model so that once exposed, individuals progress through the disease and behaviors change for both the infected and others along the way. Each agent in our model is a representative householder, whose demographic and location information is drawn from the joint distributions in the American Community Survey. This allows the researcher to get a granular understanding of the distributional equity among different demographic groups. In the epidemiological progression figure, S is susceptible, E is exposed, $P^A$ is pre-asymptomatic, $P^Y$ is pre-symptomatic, $I^A$ is asymptomatic, $I^Y$ is symptomatic, $I^H$ is hospitalized, R is recovered, and D is dead. For models details and parameter values refer to [71].

describe the role of risk tolerance in these behavioral feedbacks for each module in more detail below.

**3.2.1. Policy context.** The policy context manifests through three policy decisions related to SIP policy: timing, duration, and targeting–these aspects are discussed in greater detail below. Agents respond to the policy context by changing the degree to which they interact in professional and social spheres (Fig 1). In the professional sphere, policy determines which agents continue to interact as providers of essential services. In the social sphere, agents have the ability to determine whether or not they will comply with the SIP order by ceasing social interactions; those with the highest tolerance for risk choose not to comply with (i.e., "flout") the SIP order (Fig 2).

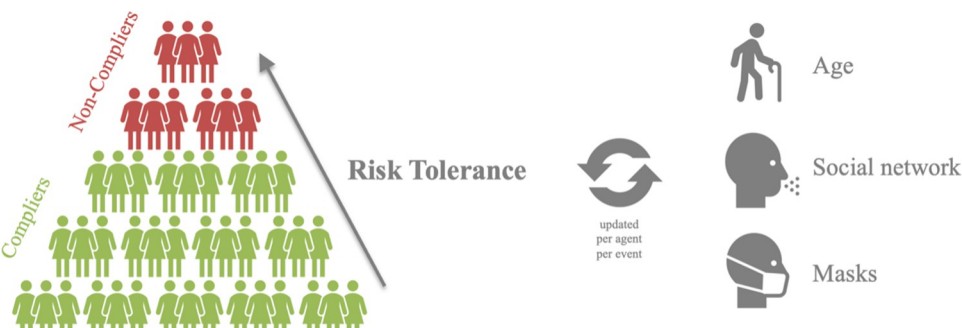

**Fig 2. Risk tolerance and protective behavior.** Agents with high risk tolerance are likely to continue to participate in the social sphere despite a shelter in place order. Agents start off with a "baseline" risk tolerance that is a decreasing function of age and update it over time as other agents in their social networks become symptomatic. In CoPE, a 25 percent non-compliance rate refers to the top 25 percent of agents who have a high tolerance to risk (rather than a random 25 percent of agents).

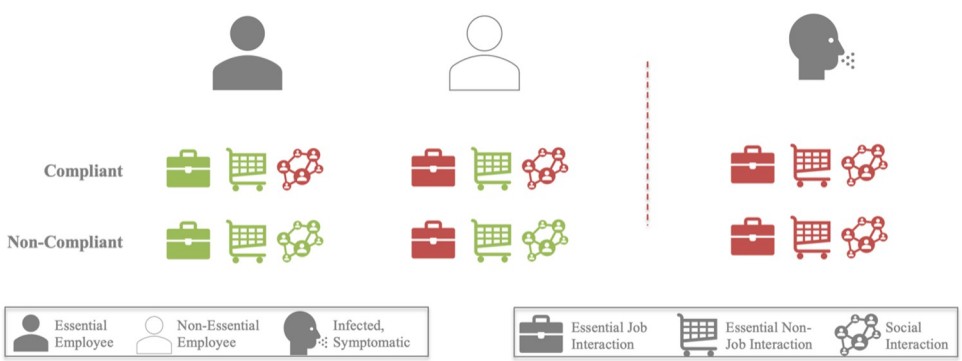

**Fig 3. Behavioral adaptation to SIP and disease.** How agents schedule their interactions is governed by a combination of the policy in effect, their individual risk tolerance, and the infection state of the agent and those in their social network at that time. When SIP is in effect, compliant individuals in essential occupations cease social interactions, and continue non-job essential interactions (as service providers). Compliant individuals in non-essential occupations cease on-the-job and social interactions but continue non-job essential interactions (as service seekers). Once individuals are infected and symptomatic, they cease all interactions.

**3.2.2. Interaction generation.** Interactions occur in both the social sphere and the professional sphere. Social interactions have the potential to expose other agents gathered at the same social "event." Professional interactions take two forms: service provision and service receipt. Service provision potentially exposes other agents within the same profession, while service receipt potentially exposes those seeking services from a profession and the agent providing that service. Within any interaction, agents update their risk tolerance to be closer to the group mean of all participants in that interaction to reflect the influence of local social norms (Fig 2). Equations governing the risk tolerance updates and asymmetric probabilities of infection are detailed in S1 Appendix.

**3.2.3. Epidemiological progression.** Each agent's progression through the disease is charted based on a compartmental epidemiological model (Fig 1, S1 Appendix) [71]. Agents then change their activities based on their disease status (Fig 3). Asymptomatically infected agents make no changes. Symptomatic agents forego any social interactions that they might otherwise have scheduled but continue to receive essential services. Hospitalized agents cease all interaction in both social and professional spheres. Recovered agents resume the activities that they engaged in pre-infection. Agents are aware of symptomatic agents in their social networks and revise their risk tolerance down (i.e., become more risk averse) as this proportion grows (Fig 2).

## 3.3. Calibration and validation

CoPE is calibrated using data and analyses pertaining to demographics, human behavior, and COVID-19. Both the agents themselves (input data) and the model components (operational modules) are empirically grounded. Agents are assigned an age, income, race/ethnicity, occupation category, and household size which is drawn from ACS5 data. The underlying distributions are geographically specific and resolved at the census block-group level. In addition, the rate at which agents participate in essential and nonessential activities is based on recent U.S. survey data [72]. Social networks–wherein nonessential activities occur–have a small world structure which has been shown to approximate human social networks[73]. In the professional sphere, agents interact based on their empirically grounded occupation category. Once infected, an agent's status (e.g., symptomatic, recovered) evolves based on COVID-19-specific disease progression data [71]. Internally, the model has face, parameter, and process validity [74].

The purpose of the CoPE tool is to demonstrate the effect of policy interventions relative to one another and to a baseline. Externally, CoPE achieves pattern validity: the pattern in the rise and fall of infections over time aligns with the observed pattern in the real world over the time period analyzed [74]. Initially, as disease spread is uncontrolled, the number of infected individuals grows rapidly and peaks before SIP policy is put in place, slowing the spread. In our baseline scenario, the first peak in hospitalization is 0.04 percent (Fig 6, center of panel E), or approximately 183 of the Travis County agents hospitalized. During the first peak, the Austin, Texas MSA (metropolitan statistical area) reported 480 hospitalizations. However, since the Austin MSA is a five-county region including Travis County the hospitalization counts are not directly comparable. Establishing point or distributional validity would more direct comparisons to real world data in a specific context, however doing so would also closely tie CoPE to that context and erode the flexibility of the tool. Pattern validity is sufficient to drive relative comparisons between scenarios.

### 3.4. Research design

While the CoPE tool can be used to study any location in the United States, for the purposes of this paper we implement CoPE to evaluate SIP policy design features in Travis County, Texas, by simulating interactions of 458,484 households over a period of 120 days from the date of initial exposure in the County. A full model run consists of 48 sample runs, each of which stochastically samples 5% of the population such that the vast majority of households are sampled at least once in the full model and most households are sampled more than once–some are included more than 10 times. The results from each sample run are aggregated into a full run accounting for the sampling procedure. For example, a household that is included in four sample runs and becomes exposed resulting in hospitalization in one of the four, but not the other three, contributes 0.25 to cumulative hospitalization in the full model.

We then simulate a baseline SIP policy scenario designed to reflect the actual SIP policy implemented in the focal area: Travis County, Texas. To compare to the baseline scenario, we then simulate a range of SIP policy configurations focusing on the timing, duration, and targeting of SIP policies, while accounting for various levels of policy compliance.

**3.4.1. Scenario development.**   Realistic baseline and alternative SIP scenarios are derived from publicly available data describing SIP orders in different states [75,76]. All of the scenarios modeled are shown in Fig 4. While most states and counties nationwide, including Travis County in Texas, imposed a SIP order around March 23, 2020 some counties in California acted a week earlier, whereas many other states such as Florida and Georgia acted a week later. Further, reports from Travis County indicated that the first known infections possibly occurred two weeks before the first cases were confirmed on March 11, 2020 [77]. As per this timeline, SIP was implemented in Travis County four weeks after the first known infection. Thus, in our baseline scenario, SIP begins 28 days after the first infections, while in the *Early* and *Late* scenarios, SIP begins one week earlier (day 21 since first infection) or later (day 35 since first confirmed case), respectively. Similarly, the shortest duration for the SIP was around 30 days as implemented in states like Georgia and Florida. Other states like Washington and New Jersey had the SIP order in place for 60 days or longer. We define the baseline SIP as 45 days in duration with *Short* and *Long* durations of 30 and 60 days respectively. As the epidemic progressed, many States went through multiple rounds of SIP orders depending on their needs. For the purpose of this study, we simulate only the first SIP order.

Essential workers–those who would continue to provide essential services even under local SIP orders–were defined in the United States early in the course of the pandemic by federal guidelines [78]. State and local governments modified the definition of essential workers depending on their contexts. For the baseline scenario, we use both federal and Texas state

**Fig 4. Visual summary of baseline and alternative SIP scenarios.**

guidelines to categorize ACS occupational classes as essential or non-essential occupations [78,79]. In the *Restrictive* essential worker scenario, only workers in *Healthcare Practitioners and Technical Occupations* continue to provide services. On the other hand, in the *Relaxed* scenario, all workers *except* those engaged in *Food Preparation and Serving Related Occupations* continue providing services during SIP. In all scenarios we assume that within our simulation timeframe, even after the SIP order is terminated, the arts and entertainment industry would not return to full normalcy, educational instruction would continue to be online, and those in computer-related occupations would primarily work from home [80]. Therefore, in our simulations those engaged in *Computer and Mathematical Occupations*, *Educational Instruction and Library Occupations* and *Arts*, *Design*, *Entertainment*, *Sports*, *and Media Occupations* curtail their professional-sphere service provision interactions even after the SIP order ends.

The SIP orders are fundamentally designed to limit social gatherings to break the chain of transmission. However, as we have noted in section 2.2, some individuals may choose to ignore the SIP guidelines and continue to engage in activities depending on their risk preferences. The level of compliance among the local population may therefore impact the outcome efficacy of the SIP orders. Therefore, to test for the sensitivity of the outcomes to compliance with the SIP order, we consider a baseline level where 75 percent of individuals with lower risk tolerance cease all non-essential activities and simulate *Low* and *High* compliance scenarios where 50 and 90 percent of the individuals respectively comply with the SIP order.

### 3.5. Analytical strategy

We analyze the results from the simulations in three stages.

**3.5.1. Isolated efficacy analysis.** In the first stage we model the changes in peak hospitalizations and overall proportion of population hospitalized under each scenario (see Fig 4) when all other factors, other than the change that defines a scenario, are held constant. For each scenario, the mean and standard deviation provide measures of effectiveness of each SIP policy-design element.

**3.5.2. Integrated efficacy analysis.** When the policy design can vary simultaneously along multiple dimensions, interactions among design elements can produce synergies or tradeoffs that increase or decrease effectiveness of the policy. Questions involving these synergies can be

difficult to assess. For example, what are the tradeoffs between a short duration SIP enacted early versus a long duration SIP enacted late? How do these impacts vary when the compliance level of the population changes? With increasing complexity of policy design and uncertainties in policy compliance, anticipating the outcomes become less straightforward.

To investigate these tradeoffs, in stage two of our analysis we simulate the incremental and interactional effect of timing and duration of SIP under the three targeting and compliance scenarios respectively (here on referred to as the *integrated scenarios*). For each interaction of targeting and compliance scenarios, we simulate a series of models randomly varying the length–in numbers of days–of the delay in implementation of SIP and the duration of SIP. The resulting models span an outcome space that is centered on the baseline and allows us to capture nuances in potential synergies or tradeoffs between different SIP policy design criteria.

**3.5.3. Equity analysis.** Both the isolated and integrated efficacy analyses focus on the overall effectiveness of policy design. ABMs allow for the further decomposition of outcomes–beyond effectiveness at the system-level–and into equity by a variety of agent characteristics. For example, the models described here track exactly which agents experience hospitalization and when, along with the agent's demographic information such as income and age. In stage three of our analysis, we use this information to analyze the distributional impacts of different policy configurations. Specifically, we focus on the *Early* and *Late* timing scenarios and decompose the daily proportion of population hospitalized by the agent's income class to reveal any distributional impacts of the SIP order, i.e., the difference in the rates of hospitalizations among agents belonging to different income classes.

## 3.6. Limitations

CoPE has several important limitations. First, while the parameters used to initialize the models are grounded in individual-level empirical data, the results in the baseline scenario are empirically validated at the pattern level as opposed to the point level [74]. For any computational model, it is important to ensure that the validation of the model is performed at the appropriate level and for the intended purpose of analysis [81]. The objective of our model is to demonstrate the relative differences between different policy and behavioral configurations, rather than accurately forecasting the real-world estimates of a particular configuration. Therefore, we focus on establishing the pattern validity of our model and pay less attention to point validity [74]. Since these patterns emerge from heterogeneity embedded in the structure of the empirical inputs, which we do not change, the relative differences in patterns are likely to be stable and informative.

A second limitation is related to the computationally intensive nature of ABMs. For a single scenario at population scale, we simulate a random sample of 458,484 agents over a series of 48 model runs. Statistically, this implies that the estimates produced by the model may not be robust at the agent level. However, the focus of our analysis is not on individual-level outcomes, but rather on aggregated, emergent, system-level outcomes [74]. At the system level (in our case the County level), the estimates are robust.

CoPE also is limited in that not all types of real-world interactions are included. We focus on professional and social interactions, which have consistently been reported as being responsible for driving the infections. Other types of interactions, for example random encounters on the street, interactions during transit, interactions at specific locations such as schools, have not been explicitly included as separate interaction generating spheres. However, these interactions are implicitly modeled to the extent that they occur in the professional or social spheres. For example, interactions at school maybe captured by the agents in the education profession, and random encounters maybe captured as part of the social interactions.

### 3.7. Comparison with other ABMs

In this section, we compare CoPE with three other recent ABMs–the Covasim model developed by Kerr et al. [82], the TRACE model developed by Hammond et al. [66] and the stochastic ABM developed by Hoertel et al. [83]–highlighting novelties and limitations of CoPE. Two novel features of CoPE as compared to the other three models relate to the treatment of policy compliance and the focus on distributional impacts. First, CoPE models individuals' adherence to policy directives as a *dynamic* process where individuals update their risk tolerances daily as a result of their infection state and that of others in their network, and base their decision to comply with the shelter in place policy on their dynamic risk tolerance value. While acknowledging its importance, neither Kerr et al. [82] nor Hoertel et al. [83] explicitly model policy compliance, whereas Hammond et al. [66] model static bounds for policy adherence. The second novelty of CoPE is its analytical focus on the equity of policy outcomes in addition to overall efficacy. We believe one of the key strengths of an ABM is being able to decompose the results based on the characteristics of the agents. Using the age-race/ethnicity-occupation-income joint distributions we obtain from the census data, we are able to develop insights into how the overall changes in policy outcomes, such as hospitalizations, differentially affect vulnerable population groups. None of the other three models provide insights on the equity of outcomes.

One limitation of CoPE, in comparison to the other models, is the focus on a single type of intervention–SIP policy, whereas the Covasim and TRACE models include the ability to model different interventions [66,82]. Additional intervention types can be built into CoPE in the future, expanding the potential for analyses of both efficacy and equity. Highlighting the comparative novelties and limitations of CoPE underscores the value of applying different approaches to provide unique insights into different dimensions of these complex problems. In the following section, we discuss the insights into efficacy and equity that CoPE can provide.

## 4. Results and discussion

### 4.1. Isolated efficacy analysis

Fig 5 presents the result of the analysis described in section 3.4.1, wherein only one SIP design criteria is varied while holding everything else constant at the baseline scenario values mentioned in Fig 4. For example, in the early and late timing scenarios, only the timing variable is changed, while holding duration, compliance and targeting variables constant at the baseline values. As expected, compared to the baseline scenario, the percentage of population hospitalized is higher when (a) the SIP is implemented late, (b) the duration of SIP is short, (c) the compliance of the population is low, or (d) the SIP is not targeted to a large enough population (IE, it is not restrictive enough). Conversely, compared to the baseline scenario the percentage of population hospitalized is lower when (a) the SIP is implemented early, (b) the duration of SIP is longer, (c) the compliance of the population is high, or (d) the SIP is targeted broadly. While these results are in expected directions, the relative magnitude of change among the different scenarios make it clear that *timing of the SIP intervention plays the most important role in managing the rate of hospitalization* assuming everything else stays the same.

In our model, one-week delay in implementing the SIP increases the percentage of population hospitalized by 81 percent (left y-axis, Fig 5) and increases the peak percentage of population hospitalized by 117 percent (right y-axis, Fig 5). Conversely, enacting the SIP one week early reduces the percentage of population hospitalized by 62 percent, and reduces the peak percentage of population hospitalized by 82 percent. This finding is substantively consistent

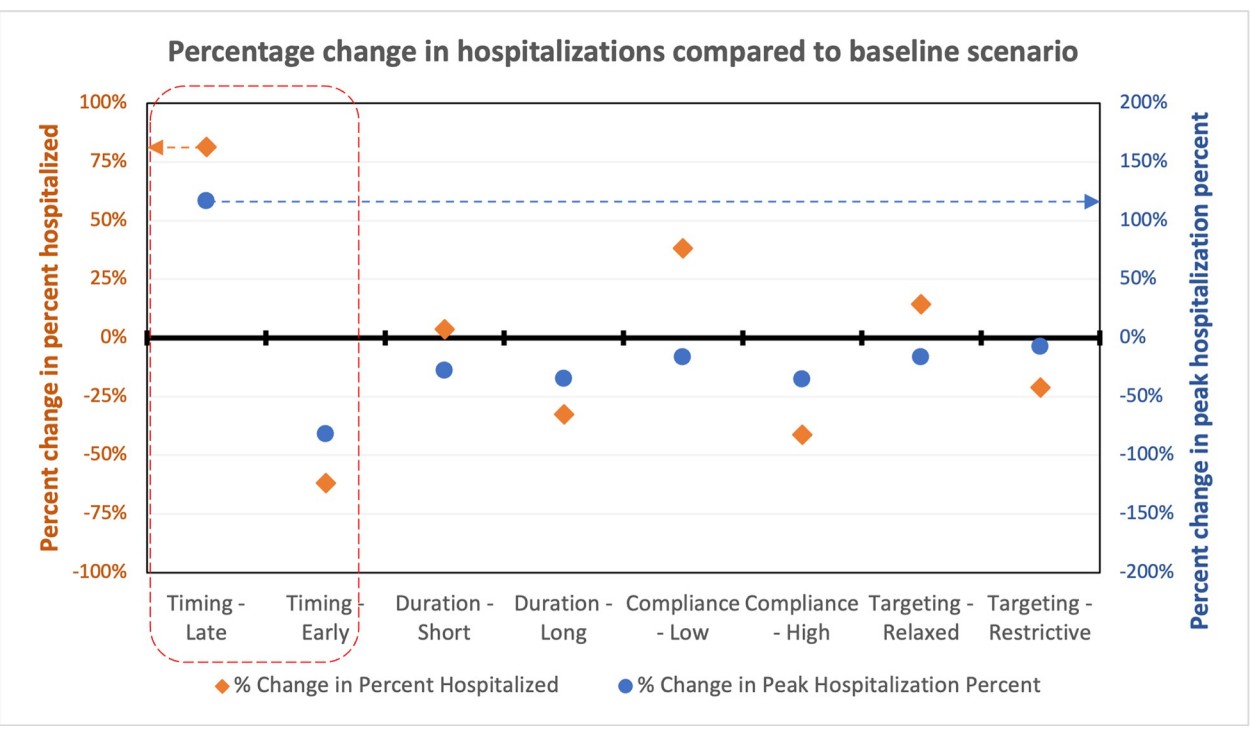

**Fig 5. Percentage change in hospitalizations in each individual scenario compared to the baseline scenario.** The y-axis on the left represents the percentage change in the percent hospitalized compared to the baseline scenario and corresponds to the orange dots in the plot. The y-axis on the right represents the percent change in peak hospitalization percent compared to the baseline scenario and corresponds to the blue dots in the plot. The black solid line along 0% represents the percentage of population hospitalized in the baseline scenario.

with previous empirical and model based studies [30–32], and highlights the need for decision-makers to act quickly to minimize the damage from shock events. Early action is only possible when there is rigorous surveillance that allows for decisionmakers to spot the early signs of emerging or re-emerging threats. Integration of non-traditional information sources into the surveillance infrastructure [84], detection of SARS-CoV-2 virus in sewage systems [85,86], coupling of real-time pathogen genomic diagnostics with epidemiology [87] and other recent advances using big data [7] highlight the evolving frontier of disease surveillance that can in turn facilitate early policy action.

### 4.2. Integrated efficacy analysis

The integrated scenarios, shown in Fig 6, also reveal the substantial contribution of SIP timing in reducing the percentage hospitalized. This analysis shows that SIP timing (early vs. late) is important in all scenarios, and the prominence of near-vertical contour lines in most scenarios suggest that the duration of the SIP becomes important only in low compliance scenarios with greater effects when the SIP is timed early, but doesn't have a notable effect in high compliance scenarios. When compliance is low, there is a pronounced non-linear relationship between timing and duration that "activates" the duration lever (see panels G,H,I in Fig 6): when the SIP is implemented early, longer durations can improve the overall effectiveness of the SIP. The sloped contour lines (e.g., panel I) also indicate that even if there is a delay in enacting the SIP, increasing the SIP duration can make it just as effective as a shorter duration SIP that could have been implemented earlier.

## Variation of total percent of population hospitalized under a range of Shelter-in-place design configurations

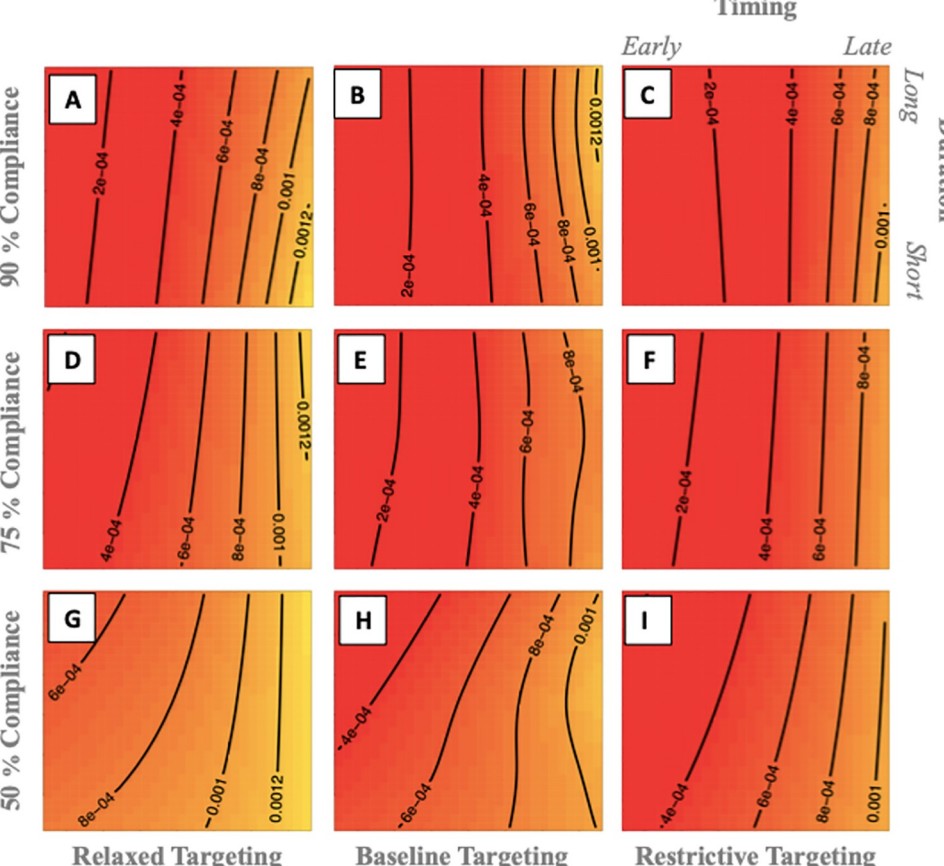

**Fig 6. Variation of total percent of population hospitalized under a range of SIP design configurations.** The three panels along the x-axis represent increasing level of SIP targeting, and the three panels along the y-axis represent increasing level of compliance. In each panel, the x-axis represents the incremental delay in timing of the SIP, and the y-axis represents the incremental increase in the duration of SIP. The variation in percentage of population hospitalized under each scenario is represented by the color gradient, where red indicates a lower percentage hospitalized and yellow indicates a higher percentage hospitalized. The black lines show the contours of equal percentage of hospitalization.

On the other hand, when the population is generally highly compliant (panels A,B,C in Fig 6), then the effectiveness of the SIP depends mostly on the timing and very little on duration. In such scenarios, no matter when the SIP is implemented, similar effectiveness is achieved with a short duration policy compared to a long duration policy. This finding has important downstream implications for reducing the economic burden from lockdowns. If the population has high levels of trust between the government and the citizens which fosters better compliance with government demands [88], then it might be possible to avoid the economic costs of extended lockdowns by enacting a short duration SIP policy. There is however a caveat to this relationship. Even when the population is highly compliant, the effectiveness of a SIP can be reduced if is inadequately targeted and subject to long delays (see panel A in Fig 6). This characterizes the Swedish experience during the first wave of COVID-19 pandemic, in which reliance on voluntary efforts to reduce exposure and much delayed SIP orders largely failed to curb transmission [89].

Finally, a shorter duration SIP that is timed late is the least effective policy in any scenario, amounting to closing the barn door after the horses have gone. Overall, these dynamics show that while the timing of the policy intervention is the most important criteria in determining the effectiveness of the policy outcome, the effectiveness is moderated in non-trivial and non-linear ways by other policy design criteria such as duration, targeting, and compliance.

### 4.3. Equity analysis

Panel A in Fig 7 shows that early SIP implementation reduces overall hospitalizations during the duration of the SIP order, but there is a resurgence once the SIP is lifted. In contrast, the disease runs through the population unchecked when the SIP is implemented late, resulting in a single peak higher hospitalization (panel C in Fig 7). While early SIP timing is effective in reducing hospitalization across the entire simulated system, considering the system as a whole ignores distributional impacts and potential inequities that can arise despite increased effectiveness along the hospitalization front.

These simulations show a tradeoff between effectiveness and equity in the distributional impact of SIP timing across different income groups. In the early timing scenario, hospitalizations among individuals in the lowest income group are disproportionately high in the second peak (panel B in Fig 7). Around day 80 in the simulation, close to 45 percent of those who are hospitalized belong to the lowest income group, who only comprise 22 percent of the population. This suggests that, despite lower prevalence of hospitalization in the system, the burden of hospitalization that remains is disproportionately borne by those in the lowest income group. In the late timing scenario, we observe a less substantially disproportionate impact on the lowest income group (panel D in Fig 7) in conjunction with *higher* overall prevalence of hospitalization.

These disproportionate impacts emerge from systemic inequality in the structure of employment, age, and income embedded in the ACS data the defines the population. Recent studies have shown that lower income and other vulnerable groups make up a higher proportion of the essential workforce and these groups may be unable to comply with social distancing guidelines. Furthermore, their nature of work in high density work-places results in the infections getting bottle-necked in these communities, leading to higher rates of infection and subsequent hospitalizations among these groups [37,42].

## 5. Conclusions

In this paper we have developed and demonstrated the application of an empirically-grounded agent-based model for *ex ante* evaluation of policy and behavioral responses to shock events, within the context of SIP policies for curbing the spread of COVID-19. We make four practical contributions to the modeling and design of policy response to shock events such as the spread of an epidemic. First, we model policy compliance as a dynamic and responsive phenomenon. We therefore more closely model the real-world dynamic responses of individuals to policy implementation. Second, consistent with previous research, we find that an early response can be very effective for avoiding deleterious effects. This highlights the importance of surveillance, rapidly implementable prepared responses, communication, and community trust in policymakers. Third, we show that the impact of timing is non-trivially moderated by other policy design aspects that can be changed in the short term such as duration and targeting, as well as societal aspects that are less pliable in the short run, such as the level of trust in governmental or health entities and compliance among the population. Even with early implementation, when compliance is low, a more restrictive policy with longer duration covering a wider population may be necessary. Finally, we show that even policy options that can be more effective

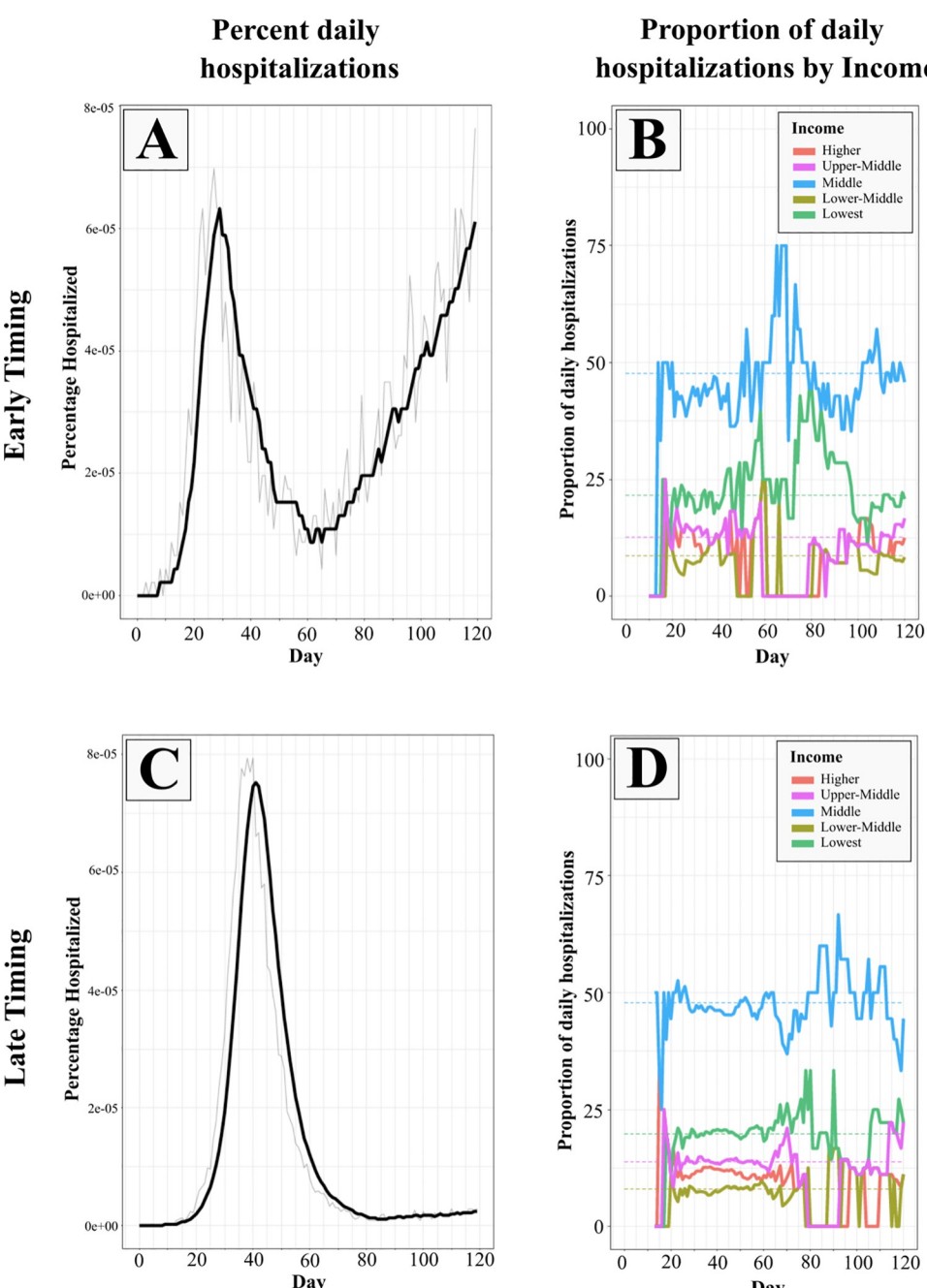

**Fig 7. Percent daily hospitalizations and proportion of daily hospitalizations by income in early and late timing scenarios.** Panels A and C show the percent of population hospitalized daily in the early and late scenarios respectively. The grey line shows the daily percentage and the solid black line shows a 7-day moving average. The solid-colored lines in Panels B and D show the proportion of daily hospitalized broken down by the agents' income categories in early and late timing scenarios respectively. The dashed colored lines represent the base percent of population in each income category. When the solid line is above the dashed line for a particular income category on a particular day, it indicates a disproportionately higher proportion of hospitalizations faced by that income category on that day.

overall can have disproportionate impacts on vulnerable populations, and even while it is impossible to precisely predict the trajectory of disease progression *it is possible to anticipate these high-level trends* if the relevant characteristics and interactive dynamics of the relevant

population are well resolved, something we do in this paper using publicly available data. Insight into the distributional-equity impacts of policy design features can help decision-makers to put in place additional welfare measures to protect the most vulnerable communities. With the CoPE modeling framework, we provide an additional tool for policymakers to use in the design of targeted strategies supporting disproportionately affected populations.

## Supporting information

**S1 Appendix. Model parameters and equations.**
(DOCX)

## Author Contributions

**Conceptualization:** Vivek Shastry, D. Cale Reeves, Nicholas Willems, Varun Rai.

**Data curation:** D. Cale Reeves, Nicholas Willems.

**Formal analysis:** Vivek Shastry, D. Cale Reeves, Nicholas Willems, Varun Rai.

**Methodology:** Vivek Shastry, D. Cale Reeves, Nicholas Willems, Varun Rai.

**Software:** Vivek Shastry, D. Cale Reeves, Nicholas Willems.

**Supervision:** Varun Rai.

**Validation:** Nicholas Willems.

**Visualization:** Vivek Shastry.

**Writing – original draft:** Vivek Shastry.

**Writing – review & editing:** Vivek Shastry, D. Cale Reeves, Nicholas Willems, Varun Rai.

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
