## [Decision Letter · Decision Letter 0]

1 Nov 2021

PONE-D-21-30912Policy and behavioral response to shock events: An agent-based model of the effectiveness and equity of policy design featuresPLOS ONE

Dear Dr. Shastry,

Thank you for submitting your manuscript to PLOS ONE. After careful consideration, we feel that it has merit but does not fully meet PLOS ONE’s publication criteria as it currently stands. Therefore, we invite you to submit a revised version of the manuscript that addresses the points raised during the review process.The manuscript requires further revisions with reference to study novelty and originality, introductory section extension, discussion of the model design and outcomes, as well as policy implications. At the same time, the paper requires serious corrections with reference to English language.

We look forward to receiving your revised manuscript.

Kind regards,

Stefan Cristian Gherghina, PhD. Habil.

Academic Editor

PLOS ONE

Journal Requirements:

Reviewers' comments:

Reviewer's Responses to Questions

**Comments to the Author**

1. Is the manuscript technically sound, and do the data support the conclusions?

Reviewer #1: Partly

Reviewer #2: Yes

2. Has the statistical analysis been performed appropriately and rigorously? 

Reviewer #1: I Don't Know

Reviewer #2: Yes

3. Have the authors made all data underlying the findings in their manuscript fully available?

Reviewer #1: Yes

Reviewer #2: Yes

4. Is the manuscript presented in an intelligible fashion and written in standard English?

Reviewer #1: Yes

Reviewer #2: Yes

5. Review Comments to the Author

Reviewer #1: This paper presents an agent-based model for evaluation of policy responses’ effectiveness in avoiding Covid-19 spread. The paper describes the operation modules underling the model. The paper does not describe the details of calibration methods. Neither does it present an evaluation of validation nor goodness-of-fit for the model.

The model code and user guides are publicly available on a hub which is addressed in the paper. The model is implemented in python and R, both of which are open source.

The data used in the model are public, however I did not find in the paper where to find it. There is a fold in the cited hub named “inputs” where I believe the data can be found. However, I was unable to read the archives inside, which may be due to operational system issues.

The paper is clear and easy to read. However, I did not feel comfortable saying the techniques and estimation are appropriate nor the conclusions are supported by the data since I did not find enough information in the paper to make possible such evaluations.

Specific comments:

1. I find the points below relevant, and it would be interesting to discuss them in the paper regarding the applicability of the model.

1.1 The model is implemented to evaluate SIP (lockdown) policy design features in Travis County, Texas, and the authors claim that it can be applied for “any location in the USA”. What does it mean location here? Only counties? Or it could be states? What the authors could say about the model being used in other countries? Or metropolitan regions? The last one is very important to evaluate policy responses for Covid-19 due to the interaction among population from different cities, including for working purposes.

1.2 Localities may differ significantly regarding several aspects such as economic sectors composition, economic and social development, population age, and culture. It will be interesting to see a results comparison between counties with different features, such as economic development.

2. What are the innovation/originality of the model compared to the model presented in Kerr Et Al (2021) and Hoertel (2020)?

3. To improve the potential of your paper, I believe it will be interesting to add some discussion about your model and the Covasim model (Kerr Et Al, 2021), since it has been used to advise policy makers in multiple countries.

4. About the results for Travis, in Figure 6 the numbers are percentages or decimals? For instance, does 0.0012 mean 0.12% or 0.0012%? A comparison between the numbers provided by the model and real data would be interesting. For instance, with 90% compliance, short duration, and early timing, the model predicts X people hospitalized. In a similar situation there were Y people hospitalized.

5. A minor point, PA is defined as pre symptomatic in the caption of Figure 1. It’s actually asymptomatic. There are some typos in Figure 1 regarding subscript instead of superscript in the variable names.

Reviewer #2: The paper “Policy and behavioral response to shock events: An agent-based model of the effectiveness and equity of policy design features” develops an agent-based model that allows for the evaluation of shelter-in-place (SIP) policies designed against the COVID-19 pandemic. The authors show, for instance, that the impact of timing of the SIP policy is moderated by other aspects of policy design and societal aspects. In my opinion, the paper makes a very important contribution on a relevant subject employing solid methods. For this reason, it deserves publication. However, the authors should perform some minor changes in the paper, as I list below:

• The Introduction of the paper is too short. The authors should extend it, elaborating on the purpose of the paper and the methods employed, as well as anticipating some results. The “COVID-19 Policy Evaluation” (CoPE), mentioned for the first time on page 11, should be briefly described in the introduction and mentioned in the abstract.

• On page 8, the expression “an SIP” should be replaced by “a SIP”.

• A more formal description of the model (CoPE), including equations and parameters, should be provided by the authors. It could be done in a separate section or an appendix.

• I suggest to the authors provide a summary table displaying all the different configurations of the SIP policy to be tested. For instance, duration: baseline (45 days), short (30 days), and long (60 days). This would improve the comprehensibility of the paper.

• On page 13, I suppose “Figure 2” should be replaced by “Figure 1”.

• On page 13, in the caption of Figure 1, both PA and PY stand for “pre symptomatic”.

• On page 14, the expression “within in the same profession” should be replaced by “within the same profession”.

• Throughout the paper, the authors use both “shelter-in-place” and “shelter in place”. Please choose just one form.

• On page 14, I suppose “Figure 2” should be replaced by “Figure 3”.

• On page 15, in the caption of Figure 3, “Compliant” should be replaced by “compliant”.

6. PLOS authors have the option to publish the peer review history of their article (what does this mean?). If published, this will include your full peer review and any attached files.

Reviewer #1: No

Reviewer #2: No

---

## [Author Response · Author response to Decision Letter 0]

6 Dec 2021

Response to the Editor

Dear Editor,

We take this opportunity to thank you and the two reviewers for the suggestions on improving this manuscript. We have addressed all of the reviewers’ suggestions, with key revisions on the following points.

• Expanded introduction section where we introduce the CoPE tool, state the objective of our study, and preview the anticipated results. 

• New sub-section 3.3 where we clarify aspects related to the calibration and validation of the CoPE tool.

• New sub-section 3.7 in which we compare CoPE with three other recent ABMs related to COVID-19.

• Supplementary information file with more information on the parameters and equations utilized within CoPE tool.

The suggestions from the reviewers helped strengthen our manuscript by providing more clarity on certain aspects of the CoPE tool while also highlighting its comparative strengths and limitations. The changes made to the manuscript does not alter any of our findings and conclusions.

Our itemized responses to each of the reviewers’ questions are detailed below.

Response to Reviewers

Please note that the comments from the reviewers are in black, and our itemized responses are in blue. 

Reviewer #1: 

This paper presents an agent-based model for evaluation of policy responses’ effectiveness in avoiding Covid-19 spread. The paper describes the operation modules underling the model. The paper does not describe the details of calibration methods. Neither does it present an evaluation of validation nor goodness-of-fit for the model.

We thank the reviewer for these observations. We have now added a separate section (section 3.4 in the revised manuscript) on calibration and validation. Our model is calibrated to empirical data to the extent that relevant data is available. The agent characteristics are derived from US census data, the interaction probabilities are based on data from the American Time Use Survey, and the COVID-19 disease progression dynamics are calibrated to parameters specified in literature (Tec et al 2020). The conclusions drawn in this study are based on the relative changes in trends between scenarios, and not based on predicted point estimates. Therefore, we focus on establishing the pattern validity of the model. This discussion has been included in the new section 3.3, as well as in the limitations section 3.6.

The model code and user guides are publicly available on a hub which is addressed in the paper. The model is implemented in python and R, both of which are open source.

The data used in the model are public, however I did not find in the paper where to find it. There is a fold in the cited hub named “inputs” where I believe the data can be found. However, I was unable to read the archives inside, which may be due to operational system issues.

We thank the reviewer for these observations regarding data availability. In order to ensure the replicability of the model, most of the input files used in the model are dynamically sourced in each run directly from the primary sources through APIs (for example, the US census and Time use survey datasets). Therefore, while some of the input files may not be available in the folders, the codes to extract them are available.

The paper is clear and easy to read. However, I did not feel comfortable saying the techniques and estimation are appropriate nor the conclusions are supported by the data since I did not find enough information in the paper to make possible such evaluations.

We thank the reviewer for raising this concern. As we have detailed in our responses to other questions raised by the reviewer, we have added new text in the manuscript explaining our calibration and validation methods. We have also included a supplementary information file with more information on the parameterization and equations used in the model. We believe the updated manuscript clarifies the concerns raised by the reviewer by more concretely establishing the linkages between data, calibration, modeling technique, validation, analytical technique, and findings and we thank them for their suggestions.

Specific comments:

1. I find the points below relevant, and it would be interesting to discuss them in the paper regarding the applicability of the model.

1.1 The model is implemented to evaluate SIP (lockdown) policy design features in Travis County, Texas, and the authors claim that it can be applied for “any location in the USA”. What does it mean location here? Only counties? Or it could be states? What the authors could say about the model being used in other countries? Or metropolitan regions? The last one is very important to evaluate policy responses for Covid-19 due to the interaction among population from different cities, including for working purposes.

We thank the reviewer for these important questions. CoPE uses US census data to generate the distributions for household characteristics such as age, income, ethnicity and occupation. Details about this process are mentioned in section 3.1 in the manuscript. The lowest resolution that this data is available in the American Community Survey is at a block-group level. Therefore, we generate the household distributions at a block-group level, and for the purpose of our analysis presented in this paper, we generate household distributions for all block-groups in Travis County. The source code takes the census code for any geographic unit in the US as an input parameter and runs the model using the household distributions for all block groups in that geographic unit. Therefore, researchers who are interested in running this analysis at the level of cities, metropolitan regions or states will be able to do so. We have incorporated this discussion into section 3.1 in the revised manuscript. Application of the model to other countries, however, requires the availability of analogous data and calibration of the model to different data sources, which may be possible but likely requires non-trivial modifications to our source code. 

1.2 Localities may differ significantly regarding several aspects such as economic sectors composition, economic and social development, population age, and culture. It will be interesting to see a results comparison between counties with different features, such as economic development.

We agree with the reviewer that localities differ significantly with regard to the mentioned characteristics. In fact, we believe that is one of the keys strengths of the CoPE model compared to ABMs that use random populations, since our agents are drawn from population models generated from block-group level household distribution data. This ensures that distribution of agents in the model mirror the demographic and occupational characteristics of the US geographic unit under analysis. This is important particularly for analyzing the equity implications of policy designs, as we have mentioned in the first paragraph of section 3.1. Our focus in this paper is to use Travis Country as a case study to demonstrate the application of the model and the types of novel outputs that it can generate. We agree with the reviewer that a comparative study of different counties would yield interesting insights – particularly related to recalibrating the model to the new empirical context. We believe this logical extension of our paper is an avenue for future research, but outside the scope of this paper. 

2. What are the innovation/originality of the model compared to the model presented in Kerr Et Al (2021) and Hoertel (2020)?

Please see our response under point #3 below.

3. To improve the potential of your paper, I believe it will be interesting to add some discussion about your model and the Covasim model (Kerr Et Al, 2021), since it has been used to advise policy makers in multiple countries.

We thank the reviewer for pointing us to the work by Kerr et al (2021) and Hoertel et al (2020). We also add to this comparison the TRACE model developed by Hammond (2020). We note two novel features of CoPE as compared to the other three models. First, CoPE models the individuals’ adherence to policy directives as a dynamic process where individuals update their risk tolerances daily as a result of their infection state and that of others in their network and base their decision to comply with the shelter in place policy on their dynamic risk tolerance value. While acknowledging its importance, neither Kerr et al (2021) nor Hoertel (2020) explicitly model policy compliance, whereas Hammond (2020) models static bounds for policy adherence. The second novelty of CoPE is its analytical focus on the equity of policy outcomes in addition to overall efficacy. We believe one of the key strengths of an ABM is being able to decompose the results based on the characteristics of the agents. Using the income-race-occupation joint distributions we obtain from the census data, we are able to develop insights into how the overall changes in policy outcomes, such as hospitalizations, differentially affect vulnerable population groups. None of the other three models provide insights on the equity of outcomes. CoPE does have one limitation compared to the other models. We focus strictly on one intervention – a shelter in place policy – to demonstrate the capability of CoPE, whereas the Covasim model (Kerr et al, 2021) and the TRACE model include the ability to model different interventions. These interventions can be built into CoPE in the future. Having highlighted the comparative novelties and limitations of CoPE, we believe different types of modeling approaches provide unique insights into different dimensions of these complex problems. We have included this discussion as a new section (3.7) in the revised manuscript.

4. About the results for Travis, in Figure 6 the numbers are percentages or decimals? For instance, does 0.0012 mean 0.12% or 0.0012%? A comparison between the numbers provided by the model and real data would be interesting. For instance, with 90% compliance, short duration, and early timing, the model predicts X people hospitalized. In a similar situation there were Y people hospitalized.

We thank the reviewer for these suggestions. The number 0.0012 means 0.12%. In our baseline scenario, we assume 75% as the compliance rate. Corresponding to this assumption, the percentage hospitalized predicted by our model is 0.0004 (figure 6, center of panel E), which for a population of 458,484 in Travis County equates to 184 agents hospitalized. During the first peak, there were 480 hospitalizations reported in the Austin MSA region. Austin MSA (metropolitan statistical area) is a five-county region, and therefore the predicted hospitalizations from CoPE for the Travis County region are comparable to the real data for the Austin MSA region. This discussion has been included under section 3.3 on page 16 in the revised manuscript.

5. A minor point, PA is defined as pre symptomatic in the caption of Figure 1. It’s actually asymptomatic. There are some typos in Figure 1 regarding subscript instead of superscript in the variable names.

The definition of PA has been corrected to pre-asymptomatic.

Reviewer #2: 

The paper “Policy and behavioral response to shock events: An agent-based model of the effectiveness and equity of policy design features” develops an agent-based model that allows for the evaluation of shelter-in-place (SIP) policies designed against the COVID-19 pandemic. The authors show, for instance, that the impact of timing of the SIP policy is moderated by other aspects of policy design and societal aspects. In my opinion, the paper makes a very important contribution on a relevant subject employing solid methods. For this reason, it deserves publication. 

We thank the reviewer for their positive response.

However, the authors should perform some minor changes in the paper, as I list below:

1. The Introduction of the paper is too short. The authors should extend it, elaborating on the purpose of the paper and the methods employed, as well as anticipating some results. The “COVID-19 Policy Evaluation” (CoPE), mentioned for the first time on page 11, should be briefly described in the introduction and mentioned in the abstract.

We thank the reviewer for these suggestions. We have expanded the introduction (on page 4) to introduce the CoPE tool, the objective of the paper and a brief note about the anticipated results. CoPE has been included in the abstract as well.

2. On page 8, the expression “an SIP” should be replaced by “a SIP”.

This typo has been corrected.

A more formal description of the model (CoPE), including equations and parameters, should be provided by the authors. It could be done in a separate section or an appendix.

We thank the reviewer for these suggestions. The description of the model, parameters and the main equations have been included in the supplementary information file.

3. I suggest to the authors provide a summary table displaying all the different configurations of the SIP policy to be tested. For instance, duration: baseline (45 days), short (30 days), and long (60 days). This would improve the comprehensibility of the paper.

In leu of a table, we have indeed included a visual summary of all SIP policy configurations tests, as can be seen in Figure 4. We hope this is the type of summary that the reviewer suggested.

4. On page 13, I suppose “Figure 2” should be replaced by “Figure 1”.

We correctly refer to Figure 2 in this instance.

5. On page 13, in the caption of Figure 1, both PA and PY stand for “pre symptomatic”.

The definition of PA has been corrected to pre-asymptomatic.

6. On page 14, the expression “within in the same profession” should be replaced by “within the same profession”.

This typo has been corrected.

7. Throughout the paper, the authors use both “shelter-in-place” and “shelter in place”. Please choose just one form.

We have replaced “shelter-in-place” with “shelter in place” throughout the paper.

8. On page 14, I suppose “Figure 2” should be replaced by “Figure 3”.

We correctly refer to Figure 2 in this instance.

9. On page 15, in the caption of Figure 3, “Compliant” should be replaced by “compliant”.

This typo has been corrected.

---

## [Decision Letter · Decision Letter 1]

17 Dec 2021

Policy and behavioral response to shock events: An agent-based model of the effectiveness and equity of policy design features

PONE-D-21-30912R1

Dear Dr. Shastry,

We’re pleased to inform you that your manuscript has been judged scientifically suitable for publication and will be formally accepted for publication once it meets all outstanding technical requirements.

Kind regards,

Stefan Cristian Gherghina, PhD. Habil.

Academic Editor

PLOS ONE

Additional Editor Comments (optional):

Reviewers' comments:

Reviewer's Responses to Questions

**Comments to the Author**

1. If the authors have adequately addressed your comments raised in a previous round of review and you feel that this manuscript is now acceptable for publication, you may indicate that here to bypass the “Comments to the Author” section, enter your conflict of interest statement in the “Confidential to Editor” section, and submit your "Accept" recommendation.

Reviewer #1: All comments have been addressed

Reviewer #2: All comments have been addressed

2. Is the manuscript technically sound, and do the data support the conclusions?

Reviewer #1: Yes

Reviewer #2: Yes

3. Has the statistical analysis been performed appropriately and rigorously? 

Reviewer #1: Yes

Reviewer #2: Yes

4. Have the authors made all data underlying the findings in their manuscript fully available?

Reviewer #1: Yes

Reviewer #2: Yes

5. Is the manuscript presented in an intelligible fashion and written in standard English?

Reviewer #1: Yes

Reviewer #2: Yes

6. Review Comments to the Author

Reviewer #1: The authors provided supplementary material about the model implementation, eliminating my main concern. They also addressed all the points I raised. In my opinion, the paper deserves to be published.

Reviewer #2: (No Response)

7. PLOS authors have the option to publish the peer review history of their article (what does this mean?). If published, this will include your full peer review and any attached files.

Reviewer #1: No

Reviewer #2: No

---

## [Editor Report · Acceptance letter]

23 Dec 2021

PONE-D-21-30912R1 

Policy and behavioral response to shock events:
An agent-based model of the effectiveness and equity of policy design features 

Dear Dr. Shastry:

I'm pleased to inform you that your manuscript has been deemed suitable for publication in PLOS ONE. Congratulations! Your manuscript is now with our production department. 

Kind regards, 

on behalf of

Dr. Stefan Cristian Gherghina 

Academic Editor

PLOS ONE